# Gallium-Telluride-Based Composite as Promising Lithium Storage Material

**DOI:** 10.3390/nano12193362

**Published:** 2022-09-27

**Authors:** Vo Pham Hoang Huy, Il Tae Kim, Jaehyun Hur

**Affiliations:** Department of Chemical and Biological Engineering, Gachon University, Seongnam 13120, Gyeonggi, Korea

**Keywords:** Ga_2_Te_3_, Ga_2_Te_3_-TiO_2_-C, anodes, Li-ion batteries, lithiation/delithiation

## Abstract

Various applications of gallium telluride have been investigated, such as in optoelectronic devices, radiation detectors, solar cells, and semiconductors, owing to its unique electronic, mechanical, and structural properties. Among the various forms of gallium telluride (e.g., GaTe, Ga_3_Te_4_, Ga_2_Te_3_, and Ga_2_Te_5_), we propose a gallium (III) telluride (Ga_2_Te_3_)-based composite (Ga_2_Te_3_-TiO_2_-C) as a prospective anode for Li-ion batteries (LIBs). The lithiation/delithiation phase change mechanism of Ga_2_Te_3_ was examined. The existence of the TiO_2_-C hybrid buffering matrix improved the electrical conductivity as well as mechanical integrity of the composite anode for LIBs. Furthermore, the impact of the C concentration on the performance of Ga_2_Te_3_-TiO_2_-C was comprehensively studied through cyclic voltammetry, differential capacity analysis, and electrochemical impedance spectroscopy. The Ga_2_Te_3_-TiO_2_-C electrode showed high rate capability (capacity retention of 96% at 10 A g^−1^ relative to 0.1 A g^−1^) as well as high reversible specific capacity (769 mAh g^−1^ after 300 cycles at 100 mA g^−1^). The capacity of Ga_2_Te_3_-TiO_2_-C was enhanced by the synergistic interaction of TiO_2_ and amorphous C. It thereby outperformed the majority of the most recent Ga-based LIB electrodes. Thus, Ga_2_Te_3_-TiO_2_-C can be thought of as a prospective anode for LIBs in the future.

## 1. Introduction

In recent decades, the rapidly growing desire for portable electronics, electric vehicles, and smart grids has resulted in innovative Li-ion batteries (LIBs) with high energy densities. However, the conventional carbonaceous anodes utilized in LIB systems have low capacities and rate capabilities, making LIBs unsuitable for meeting the requirements of advanced devices. This has necessitated the discovery of new high-performance electrode materials [1,2,3,4,5,6,7]. Li alloys containing components, for instance, Ge, Si, Sb, and Sn, have been proposed as attractive alternatives to high-performance LIBs because their theoretical capacities are considerably higher (Li-Ge: 1384 mAh g^−1^, Li-Si: 3590 mAh g^−1^, Li-Sn: 993.4 mAh g^−1^, Li-Sb: 660 mAh g^−1^) than those of commercial graphite anodes (372 mAh g^−1^) [8,9,10,11,12,13,14,15,16,17,18,19,20,21,22]. However, the cycling instabilities of these alloys, which are associated with significant volume changes during Li insertion/extraction, have limited their commercialization [23,24,25,26,27,28].

With the ability to alloy with two Li-ions ((Li_2_Ga), Ga is deemed a feasible anode material for LIB. This provides theoretical Li-storage specific capacities of 769 mAh g^−1^, respectively. Furthermore, Ga anodes have high theoretical volumetric Li-storage capacities (4545 mAh cm^−3^) due to the high density of Ga (5.91 g cm^−3^ at ambient temperature) [29,30]. As a result, various Ga-based anodes have been studied; however, they generally experience liquid agglomeration during cycling because of the low melting temperature of Ga (29.8 °C), leading to low cycling performance [31,32,33,34,35,36].

Among the chalcogenide elements, S- and Se-based alloys or composite materials have been widely selected as anode materials in rechargeable LIB systems [37,38,39,40,41,42,43,44,45,46,47,48]. Te has recently been investigated as a viable electrode material for LIBs [49,50,51]. When utilized as an electrode material, Te has various advantages over other chalcogen group elements. Te possesses the highest electronic conductivity among all nonmetallic materials (approximately 2 × 10^−2^ S cm^−1^), which is significantly greater than those of S (approximately 5 × 10^−16^ S cm^−1^) and Se (approximately 1 × 10^−4^ S cm^−1^). Furthermore, Te retains a high theoretical volumetric capacity (Li: 2621 mAh cm^−3^), which is associated with its high density (6.24 g cm^−3^) [51]. However, Te cannot overcome the capacity fading attributed to the large volume variation during cycling [52,53,54,55,56,57,58,59,60,61].

Various applications of gallium telluride, which is a binary compound of Ga and Te, have been studied, such as optoelectronic devices, radiation detectors, solar cells, and semiconductors, owing to its unique electronic, mechanical, and structural properties [62,63,64,65]. Among various gallium tellurides (GaTe, Ga_3_Te_4_, Ga_2_Te_3_, and Ga_2_Te_5_), Ga_2_Te_3_ is a steady compound that is odorless, black, brittle, and non-toxic. Because Ga_2_Te_3_ has a high melting point of 789 °C, and it does not undergo Ga dissolution and agglomeration during cycling, it can be safely used as a LIB anode material [66]. In addition, the high density (5.57 cm^−3^) of Ga_2_Te_3_ allows for high theoretical volumetric capacities for LIBs (2858 mAh cm^−3^) [67]. Despite these suitable features, the application of Ga_2_Te_3_ as an LIB anode material has not been studied in detail. In addition, ordinary considerations such as unstable stability, irreversible capacity, and inferior Coulombic efficiency remain significant challenges due to the large volume expansion during electrochemical reactions. Thus, an efficient strategy is needed to achieve stable and high-performance anode materials. To this end, many approaches have been investigated to resolve the aforementioned issues. The employment of diverse carbonaceous materials (including graphite, carbon nanotubes, porous carbon, carbon black, carbon fiber, and graphene (or reduced graphene oxide)) to active materials has been demonstrated as an effective approach [68,69,70,71,72,73]. The carbon-based materials not only buffer the large volume change of active materials and prevent electrode pulverization but also enhance the electrical conductivity. Nevertheless, the presence of excess carbon concentration leads to a specific capacity reduction due to its low theoretical capacity. Another strategy is to create a composite or compound that contains passive metal elements (such as Ni, Cu, Fe, Co, V, and Mo) that are alloyed with the active material to improve its mechanical and electrical conductivity [74,75,76,77,78,79]. As a last effective strategy for preventing volume change, ceramic-based materials such as TiO_2_, TiC, Al_2_O_3_, Si_3_N_4_, and MgO are cooperated with active materials [80,81,82,83,84]. Although certain ceramics possess low specific capacities, they can prevent agglomeration and volume changes in the active material owing to their great mechanical properties.

In this work, we synthesized a Ga_2_Te_3_-based composite electrode (Ga_2_Te_3_-TiO_2_-C) using simple high-energy ball milling (HEBM) and demonstrated its suitability for LIB anodes. The feasibility of the Ga_2_Te_3_-TiO_2_-C anode for LIBs was examined by performing galvanostatic measurements, differential capacity analysis, and electrochemical impedance spectroscopy (EIS). More importantly, the Li insertion/extraction electrochemical phase-change mechanism of Ga_2_Te_3_-TiO_2_-C anodes was studied via ex situ X-ray diffraction (XRD) analysis. The optimal C concentration of the Ga_2_Te_3_-TiO_2_-C composite was determined through various electrochemical measurements of the as-prepared LIBs. Ga_2_Te_3_-TiO_2_-C (10%) exhibited high cycling and rate performances comparable to those of the most recent Ga-based electrodes.

## 2. Experimental Materials and Methods

### 2.1. Material Synthesis

Ga_2_Te_3_ was synthesized using simple HEBM, as shown in Figure 1. In the first step, a mixture of Ga_2_O_3_ (99.99%, Sigma Aldrich, St. Louis, MI, USA), Te powder (99.8%, Alfa Aesar, Haverhill, MA, USA), and Ti (325 mesh, 99.99%, Alfa Aesar), in a molecular ratio of 2:3:6 was placed in a bowl containing zirconium oxide balls. The ratio of the balls and powder mixture was 20:1. The powder compound was ball milled for 10 h at 300 rpm under an Ar atmosphere. In the second step, the obtained powder (Ga_2_Te_3_-TiO_2_) was mixed with acetylene carbon black powder (C) (99.9+%, Alfa Aesar, bulk density: 170–230 g L^−1^, S.A.: 75 m^2^ g^−1^) in mass ratios of 9:1, 8:2, and 7:3 (denoted as Ga_2_Te_3_-TiO_2_-C (10%), Ga_2_Te_3_-TiO_2_-C (20%), and Ga_2_Te_3_-TiO_2_-C (30%), respectively). These combinations were manually ground and then subjected to a 10-h ball milling process under identical conditions as the initial milling. The following is the mechanochemical reaction route for synthesizing Ga_2_Te_3_-TiO_2_-C:

First step:2Ga_2_O_3_ + 6Te + 3Ti → 2Ga_2_Te_3_ + 3TiO_2_ (Ga_2_Te_3_-TiO_2_)(1)

Second step:Ga_2_Te_3_-TiO_2_ + C → Ga_2_Te_3_-TiO_2_-C(2)

### 2.2. Material Characterization

Ga_2_Te_3_-TiO_2_ and Ga_2_Te_3_-TiO_2_-C crystal structures were determined using powder XRD (D/MAX-2200 Rigaku, Japan) with Cu Kα (λ = 1.54 Ả) radiation at a scan rate of 2° min^−1^. High-resolution transmission electron microscopy (HRTEM, JEOL JEM-2100F), scanning electron microscopy (SEM, Hitachi S4700, Japan), and energy-dispersive X-ray spectroscopy (EDXS) were employed to examine the microscopic morphology of the as-synthesized composite materials. The chemical states of the produced materials were assessed using X-ray photoelectron (XP) spectroscopy (XPS, Kratos Axis Anova). Ga_2_Te_3_-TiO_2_-C anode reaction process was investigated using ex situ XRD.

### 2.3. Electrochemical Measurements

A conventional casting technique was used to prepare all of the electrodes. Briefly, a slurry including the active material, poly (acrylic acid) (PAA, Mw 450000, Sigma Aldrich) binder, and conductive carbon (Super-P, 99.9%, Alfa Aesar) in a ratio of 7.0:1.5:1.5 (*w*/*w*) was dissolved into the N-methyl-2-pyrolidone (NMP) solution with the solid-to-liquid ratio of 1:12.5, and then casted on a Cu foil current collector. The cast electrodes were transferred to an Ar gas-filled glove box for cell assembly after being dried in a vacuum oven overnight at 70 °C to completely eliminate the solvent residue. For half-cell testing, a coin-type cell (CR2032) was utilized with Li metal foil as a counter electrode, polyethylene as a separating membrane, and 1 M LiPF_6_ in diethyl carbonate/ethylene carbonate (1:1 by *v*/*v*) as an electrolyte. Using a battery-testing device ((WBCS3000, WonATech, South Korea), the electrochemical performance of Ga_2_Te_3_-TiO_2_-C was assessed. When compared to Li/Li^+^, a 0.01 to 2.5 V voltage range was applied to establish the galvanostatic charge–discharge (GCD) profile. To describe the electrochemical responses of the electrodes with Li^+^, cyclic voltammetry (CV) analyses were conducted at a scanning rate of 0.1 mV s^−1^. A battery cycler (WBCS3000, WonATech) was used to measure the rate capability at various current densities (0.1, 0.5, 1, 3, 5, and 10 A g^−1^), and the current densities are calculated based on the per gram Ga_2_Te_3_. The EIS was conducted using a ZIVE MP1 (WonaTech) analyzer in the frequency range of 100 kHz–100 MHz at an AC amplitude of 10 mV.

## 3. Results and Discussion

The XRD pattern of the as-synthesized Ga_2_Te_3_-TiO_2_ obtained by HEBM is shown in Figure 2a. The XRD pattern was the same as that of monoclinic Ga_2_Te_3_. The peaks at 26.2°, 30.3°, 43.4°, 51.4°, 53.8°, 63.0°, 69.4°, and 79.5° corresponded to the (111), (200), (220), (311), (222), (400), (331), and (422) planes of Ga_2_Te_3_, respectively. The relatively small peaks observed at 28.6°, 33.3°, 44.5°, and 75.6° were attributed to the (002), (311), (601), and (623) planes of TiO_2_, respectively. The insignificant TiO_2_ peaks below 20° are associated with the low TiO_2_ content in the composite (as shown in Appendix A) [85,86]. The addition of amorphous C decreased the crystallinity of Ga_2_Te_3_ and TiO_2_ in Ga_2_Te_3_-TiO_2_-C (Appendix A) [87]. It was clear that the target product had been completely transformed from the raw elements by a solid-state reaction because there were no impurity peaks for the precursor components (Ga, Ti, Te or Ga_2_O_3_). The chemical bonding of Ga_2_Te_3_-TiO_2_-C (10%) was assessed using XPS (Figure 2b–g). The presence of Ga, Te, O, Ti, and C in Ga_2_Te_3_-TiO_2_-C (10%) was shown in the XPS survey spectrum in Figure 2b, along with their specific binding energies. The Ga 3d orbital level signal in Figure 2c corresponded to Ga 3d_3/2_ (20.9 eV) and Ga 3d_5/2_ (19.8 eV), whereas the peaks in Figure 2d were ascribed to Te 3d_3/2_ (583.9 eV) and Te 3d_5/2_ (573.6 eV), which confirms the formation of Ga_2_Te_3_ alloy after the HEBM process. Furthermore, the existence of Te-O bonding with signals at 576.0 and 586.4 eV (Figure 2d) on the Ga_2_Te_3_-TiO_2_-C (10%) surface implied that partial surface oxidation of active Ga_2_Te_3_ [88,89]. Although obvious oxidation is observed for Ga_2_Te_3_, the air does not seem to have too much of an effect on anode composites. Indeed, there were no impurities nor significant compositional changes for the composite anode (Appendix A) compared with the as-synthesized Ga_2_Te_3_-TiO_2_-C (10%) powder (Appendix A). In addition, oxidation was mainly observed for Te due to the Te-rich compound of Ga_2_Te_3._ As shown in Appendix A, the atomic percent of Te (27%) was greater than that of Ga (17%). Therefore, Te sites seem to be more affected by the rapid oxidation in air [90]. Ga 3D hybridization was found because of the constitution of the O 2 s peak at 23.7 eV [86,91]. Regarding the formation of TiO_2_, Ti-O binding was demonstrated through the detection of the orbital level signals of Ti 2p_3/2_ (458.9 eV) and Ti 2p_1/2_ (464.6 eV) (Figure 2e) along with the O 1 s peak (530.9 eV) (Figure 2f). More importantly, the binding energy level in the O 1 s spectrum at 532.3 eV (Figure 2f) showed the formation of hydroxide groups on the active surface of Ga_2_Te_3_, implying hydrogen bond formation with functional moieties (carboxylate functional groups) of PAA binder due to its possessing high affinity. The strong binding between the PAA binder and hydroxides on Ga_2_Te_3_ is expected to prevent particle agglomeration and maintain good contact between the current collector and electrode [86,92]. The XPS results of C 1 s in Figure 2g showed binding energies at 284.6, 285.0, and 285.9 eV, which indicate C–C, C–O–C, and C–O=C bonds, respectively. These results confirmed the constitution of the target ternary composites (Ga_2_Te_3_, TiO_2_, and C for Ga_2_Te_3_-TiO_2_-C (10%)).

Morphological and structural analyses of Ga_2_Te_3_-TiO_2_-C (10%) were investigated using SEM, HRTEM, and EDXS, as shown in Figure 3. The SEM images showed that the particle size ranged from submicrometers to a few micrometers (Figure 3a,b). The HRTEM images (Figure 3c and Appendix A) revealed crystalline lattice spacings of 0.340, 0.294, 0.208, and 0.170 nm, which corresponded to the (111), (200), (220), and (222) planes of Ga_2_Te_3_, respectively, and 0.311 nm attributed to the (002) plane of TiO_2_. Additionally, amorphous C was created as a flat surface layer around Ga_2_Te_3_ and TiO_2_, and it was anticipated to serve as a buffering matrix for the active material. In the Ga_2_Te_3_-TiO_2_-C (10%) sample, the EDXS mapping analysis of the scanning transmission electron microscopy image (Figure 3d) revealed a homogeneous dispensation of each element (Ga, Te, Ti, O, and C). Furthermore, the SEM–EDXS analysis results (Appendix A) of Ga_2_Te_3_-TiO_2_-C (10%) consistently showed that the component elements were uniformly scattered throughout the composite. Additionally, the quantitative examination of the EDS results demonstrated that the stoichiometric ratio of the component elements was nearly similar to the theoretical values.

Ga_2_T_3_-TiO_2_-C with various C content for LIBs was investigated electrochemically using half-cells electrode systems (Figure 4). The GCD voltage profiles of Ga_2_Te_3_-TiO_2_-C (10%), Ga_2_Te_3_-TiO_2_-C (20%), and Ga_2_Te_3_-TiO_2_-C (30%) are shown in Figure 4a and Appendix A. The first discharge/charge performance of Ga_2_Te_3_-TiO_2_-C (10%), Ga_2_Te_3_-TiO_2_-C (20%), and Ga_2_Te_3_-TiO_2_-C (30%) were 892/677, 837/586, and 789/568 mAh g^−1^, respectively, which corresponded to initial Coulombic efficiencies (ICEs) of 75.9%, 70.0%, and 71.9%, respectively. The three electrodes experienced irreversible capacity losses in the initial cycle that were attributed to the development of a solid electrolyte interfacial (SEI) layer. On the basis of the EDXS results (Appendix A) and the computed theoretical capacities of the separate elements (Appendix A), the capacity contributions of C and TiO_2_ to Ga_2_Te_3_-TiO_2_-C (10%) were estimated to be ~9% and ~16%, respectively. Therefore, active Ga_2_Te_3_ (75% of the total capacity) was the principal source of the capacity of the electrode. The primary role of C and TiO_2_ was as a buffering matrix (25% capacity involvement), which reduced the volume variation of the active material. Furthermore, the theoretical capacity contribution of different components in Ga_2_Te_3_-TiO_2_-C (20 and 30%) was also determined (Appendix A). Compared with Ga_2_Te_3_-TiO_2_-C (10%), increase in the C concentration results in a decrease in the capacity contribution of active material. Based on Appendix A and Appendix A, the calculated capacity contribution of the active material Ga_2_Te_3_ was 75, 61, and 53%, resulting in the actual Ga_2_Te_3_ capacity of ~576, ~401, and ~314 mAh g^−1^ for the Ga_2_Te_3_-TiO_2_-C (10%), Ga_2_Te_3_-TiO_2_-C (20%), and Ga_2_Te_3_-TiO_2_-C (30%), respectively. Notably, the measured capacities of Ga_2_Te_3_-TiO_2_-C (10%) and Ga_2_Te_3_-TiO_2_ (455 and 477 mAh g^−1^, respectively, as computed in Appendix A) that were higher than their theoretical capacities are most likely ascribed to the interfacial Li-ion storage and electrolyte decomposition. The specific performance of the lowest C content electrode (Ga_2_Te_3_-TiO_2_-C (10%)) was the highest in terms of stability and capacity. It reached 768.9 mAh g^−1^ with capacity retention (CR) of 99.8% after 300 cycles at 100 mA g^−1^ (Figure 4b). The specific capacities of Ga_2_Te_3_-TiO_2_-C (20%) and Ga_2_Te_3_-TiO_2_-C (30%) were 587.3 and 585.3 mAh g^−1^ after 300 cycles at 100 mA g^−1^, respectively, which corresponded to a CR of 89.2% and 98.7%, respectively. This behavior was further explained using Coulombic efficiency (CE, Appendix A) and differential capacity plot (DCP) analyses of the first 300 cycles (Appendix A). The CE increased gradually and steadily. Particularly, the CE achieved almost 99.13% after 150 cycles, with the possibility that side reactions were involved until this point. Then, the CE decreased slightly and stabilized at 98.5% after 300 cycles. The DCP analysis showed that the main reduction peaks (at ~1.22 and ~1.69 V) remained unchanged for 300 cycles. However, the oxidation peaks (at ~0.41, ~0.98, and ~1.25 V) were stable for 100 cycles, after which they became broader and shifted toward a high voltage. Nevertheless, this polarization had an almost negligible effect on the lithiation/delithiation, resulting in a stable capacity after 300 cycles. This was because the TiO_2_ matrix and lowest C content (10%) effectively prevented the side reactions that could result from good electrode–electrolyte contact at 100 mA g^−1^. At 500 mA g^−1^, a similar trend was observed (Figure 4c). In this instance, the performance increased until 250 cycles, then slightly decreased, and finally became saturated (~600 mAh g^−1^). The CE variation (Appendix A) and DCP analysis both showed this tendency (Appendix A). According to Appendix A, the magnitudes of the reduction (at ~0.98 and ~1.69 V) and the oxidation (at ~0.41, ~1.58, and ~1.85 V) rose for 200 cycles with a decrease in polarization and then reduced after 200 cycles with a minor increase in polarization. This was followed by a reduction in polarization after 400 cycles (Appendix A). Therefore, although the capacity decreased from 250 cycles to 400 cycles, it saturated after 400 cycles. Under a high current density, an electrode requires demanding lithiation/delithiation conditions (500 mA g^−1^). This makes it more difficult to achieve steady and stable cycling [93,94,95]. To comprehend the steady rise in the performance, the variations in the DCP curves, as a function of the cycle number, were studied at 100 and 500 mA g^−1^ (Appendix A). The DCP curves of the Ga_2_Te_3_-TiO_2_-C (10%) electrode showed that the overall intensity of the redox peaks were relatively stable as the cycle number increased at 100 mA g^−1^, indicating a stable capacity until 300 cycles. However, at 500 mA g^−1^, the overall magnitudes of the redox peaks rose with the cycle number until 300 cycles, reduced from 300 cycles to 400 cycles, and became saturated after 400 cycles. The CE variations at 100 and 500 mA g^−1^ of Ga_2_Te_3_-TiO_2_-C with varied C concentrations were compared in Appendix A. The detailed CE values for the Ga_2_Te_3_-TiO_2_-C (10%), Ga_2_Te_3_-TiO_2_-C (20%), and Ga_2_Te_3_-TiO_2_-C (30%) electrodes over the first ten cycles are described in Appendix A (at 100 mA g^−1^) and Appendix A (at 500 mA g^−1^). As displayed in Appendix A, the ICE of the Ga_2_Te_3_-TiO_2_-C (10%) electrode (75.9%) were slightly higher than those of the Ga_2_Te_3_-TiO_2_-C (20%) (ICE = 69.9%) and Ga_2_Te_3_-TiO_2_-C (30%) electrodes (ICE = 72.1%). The CE of the Ga_2_Te_3_-TiO_2_-C (10%) electrode was the highest after ten cycles. At 500 mA g^−1^, it revealed a similar tendency (Appendix A). After the first cycle, the high CE of the Ga_2_Te_3_-TiO_2_-C (10%) electrode suggested that lithiation/delithiation was highly reversible. The Ga_2_Te_3_-TiO_2_-C (10%) CV curves for the first five cycles in the voltage range of 0.01–2.5 V vs. Li/Li^+^ were shown in Figure 4d. Due to SEI layer formed on the electrode surface, the CV curve in the first cycle was noticeably different from that of the subsequent cycles. The intercalation of Li into Ga_2_Te_3_ to form Li_2_Te and Ga is indicated by a substantial reduction peak at 1.37 V in the first discharge. The peak at 0.98 V was responsible for the interaction between Ga and Li to generate Li_2_Ga. Thus, Li_2_Te and Li_2_Ga were the final products after the discharge step was completed. The three oxidation peaks were shown at 0.92, 1.56, and 1.88 V in the charge process. The first peak was caused by the complete exclusion of Li, turning Li_2_Ga into Ga. Ga began to intrude into Li_2_Te to form Ga_2_Te_3_ when the anode was charged to 1.56 and 1.88 V. In the ex situ analyses, this phase change is examined in detail. After the second cycle, the curves nearly overlapped, indicating the excellent reversibility and stability of Ga_2_Te_3_-TiO_2_-C (10%). Compared to Ga_2_Te_3_-TiO_2_-C (10%), Ga_2_Te_3_-TiO_2_-C (20%) and Ga_2_Te_3_-TiO_2_-C (30%) showed similar stability in terms of the polarization of the reduction and oxidation peaks after the second cycle (Appendix A). The control experiments of Ga_2_Te_3_-TiO_2_-C (10%) with PVDF were conducted to better define the role of the PAA binder. The oxidation occurring on the Ga_2_Te_3_ surface positively affects the electrochemical performance by stabilizing the electrode structure through hydrogen bonding between hydroxyl groups in Ga_2_Te_3_ and carboxylate groups in the PAA binder. As shown in Appendix A, the cyclic performance of the composite with PAA binder showed significantly enhanced performance compared to the composite with PVDF. Besides, CV curves do not overlap with the increase in the cycle number for the composite with PVDF, indicating the irreversible cycling behavior. This result is consistent with the previous study in which the cycling performance of oxidized active material was enhanced with PAA binder [86]. The rate performances (Figure 4e) and normalized capacity retention values (Figure 4f) of the electrodes were studied at different current densities. In Figure 4e, the average discharge capacities of Ga_2_Te_3_-TiO_2_-C (10%) were significantly greater than those of Ga_2_Te_3_-TiO_2_-C (20%) and Ga_2_Te_3_-TiO_2_-C (30%), which were 708, 706, 687, 665, 636, and 613 mAh g^−1^ at current densities of 0.1, 0.5, 1.0, 3.0, 5.0, and 10.0 A g^−1^, respectively. Surprisingly, even at a high current density of 10 A g^−1^, Ga_2_Te_3_-TiO_2_-C (10%) had capacity retention of up to 96% (Figure 4f). Furthermore, Ga_2_Te_3_-TiO_2_-C (10%) showed high rate performance when the discharge rate was reduced from 10 A g^−1^ to 0.1 A g^−1^, resulting in high-capacity retention (99%).

The phase change mechanism during the lithiation/delithiation process of the Ga_2_Te_3_-TiO_2_-C(10%) electrode was investigated using ex situ XRD (Figure 5a). Peaks corresponding to Li_2_Te and Ga were observed at a discharge voltage of 1.37 V (D-1.37 V). When the electrode was completely discharged (D-5 mV), Li_2_Ga peaks emerged and Li_2_Te peaks remained. The Li_2_Ga phase partly disappeared when the electrode was charged to 0.92 V (C-0.92 V). In the charging state at 1.56 V, the Li_2_Te phase partly disappeared, Ga was observed, and Li_2_Ga completely disappeared. Only the peaks corresponding to Ga_2_Te_3_ were observed again when the electrode was completely charged to 2.5 V (C-2.5 V). Ga_2_Te_3_ undergoes structural changes during first lithiation/delithiation as follows:

Discharging:Ga_2_Te_3_ → Li_2_Te + Ga → Li_2_Te + Li_2_Ga(3)

Charging:Li_2_Te + Li_2_Ga → Li_2_Te + Ga → Ga_2_Te_3_(4)

It is noteworthy that after the first cycle, the Ga_2_Te_3_ phase (major peaks at 51.4°, 53.8°, and 69.4°) was completely restored with no impurity peaks, showing the highly reversible interaction of Ga_2_Te_3_ with Li-ions. The active material was well shielded from pulverization and delamination because of volume expansion thanks to the strong binding between Ga_2_Te_3_ and TiO_2_-C. As schematically shown in Figure 5b, the ex situ XRD result demonstrated the alloying/dealloying and conversion mechanism of the Ga_2_Te_3_ electrode during the first charge/discharge process.

At the 1st, 5th, and 20th cycles, the EIS profiles of the Ga_2_Te_3_-TiO_2_-C (10%), Ga_2_Te_3_-TiO_2_-C (20%), and Ga_2_Te_3_-TiO_2_-C (30%) electrodes were obtained (Figure 6). The equivalent circuit to fit EIS profile shown in Figure 6d includes the SEI layer resistance (R_SEI_), charge–transfer resistance (R_ct_), electrolyte resistance (R_b_), interfacial double-layer capacitance (C_dl_), constant phase element (C_PE_), and Warburg impedance (Z_w_). R_ct_ at the electrode–electrolyte interface is shown by compressed semicircles in the mid-frequency region of the Nyquist plots. As the number of cycles grew from 1 to 20, cells containing various concentrations of C displayed decreasing sizes of semicircles, suggesting that R_ct_ decreased (Figure 6a–c). Ga_2_Te_3_-TiO_2_-C (10%) showed the lowest value of R_ct_ after 20 cycles (Appendix A), indicating the optimal charge transport circumstances, which resulted in the highest electrochemical performance.

The electrochemical Li-storage behaviors were determined from the above results. Because amorphous C was delivered as a buffer to limit volume expansion during the lithiation/delithiation process, the cyclic performance was stable. Nevertheless, the regulation of the C content played an important role. A C content of 10% was sufficient to achieve high electrochemical efficiency for the LIBs. When the C content was increased, the specific capacity was rather decreased due to the reduced active material in the composite. In addition, TiO_2_ synergistically prevented electrode pulverization and improved Li-ion diffusion. Therefore, the Ga_2_Te_3_-TiO_2_-C (10%) electrode showed high electrochemical performance and fast kinetics due to the cooperative impact of the TiO_2_-C hybrid matrix, as demonstrated in Figure 7. Accordingly, the capacity of the Ga_2_Te_3_-TiO_2_-C (10%) electrode was higher than those of recently reported Ga-based anodes for LIBs (Table 1).

## 4. Conclusions

Ga_2_Te_3_-TiO_2_-C was successfully prepared via HEBM and investigated as a propitious anode material for LIBs. The morphology, chemical state, and crystal structure of Ga_2_Te_3_-TiO_2_-C were investigated through XRD analysis, SEM, EDXS, HRTEM, and XPS. To identify the conversion/recombination reaction mechanism of the Ga_2_Te_3_ anode during the lithiation/delithiation processes, ex situ XRD analysis was studied. The major strategy for achieving high capacity and long-term cycling performance for the Ga_2_Te_3_-TiO_2_-C nanocomposite was to homogeneously embed nanoconfined Ga_2_Te_3_ crystallites within an electronically conductive TiO_2_-C matrix. This promoted Li-ion diffusion kinetics and improved the mechanical stability by accommodating the change in the volume of the Ga_2_Te_3_ particles and preventing the agglomeration of Ga. As a result, the Ga_2_Te_3_-TiO_2_-C electrode showed high rate capability (CR of 96% at 10 A g^−1^ compared to 0.1 A g^−1^), as well as great reversible specific capacity (769 mAh g^−1^ at 100 mA g^−1^ after 300 cycles). It thereby outperformed the majority of the most recent Ga-based LIB electrodes. The electrochemical performance of Ga_2_Te_3_-TiO_2_-C was enhanced by the synergistic interaction of TiO_2_ and amorphous C. Thus, Ga_2_Te_3_-TiO_2_-C can be thought of as a prospective anode material for LIBs of the future.

## Figures and Tables

**Figure 1 nanomaterials-12-03362-f001:**
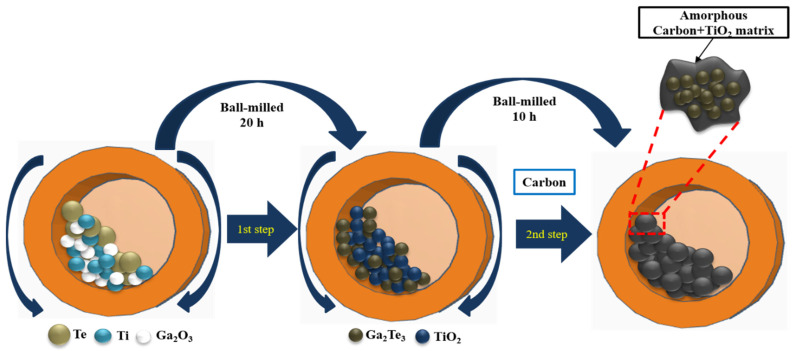
Schematic of Ga_2_Te_3_-TiO_2_-C synthesis using two-step HEBM process.

**Figure 2 nanomaterials-12-03362-f002:**
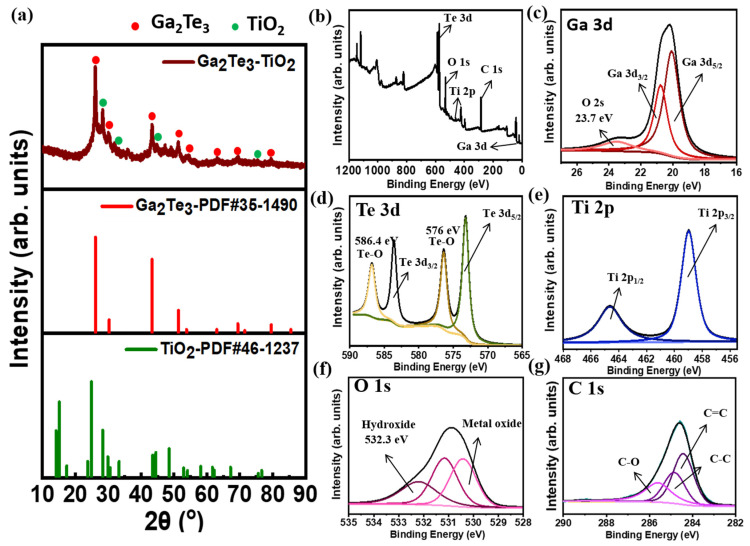
(**a**) XRD pattern of Ga_2_Te_3_-TiO_2_; (**b**) XPS survey spectrum; (**c**) high-resolution XP spectra of Ga 3d; (**d**) Te 3d; (**e**) Ti 2p; (**f**) O 1 s and (**g**) C 1 s for Ga_2_Te_3_-TiO_2_-C (10%).

**Figure 3 nanomaterials-12-03362-f003:**
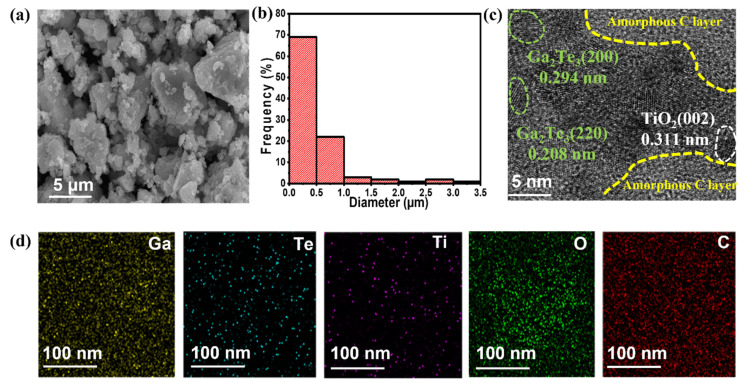
(**a**) SEM image; (**b**) particle size distribution; (**c**) HRTEM image and (**d**) EDXS elemental mappings of Ga, Te, Ti, O, and C for Ga_2_Te_3_-TiO_2_-C (10%).

**Figure 4 nanomaterials-12-03362-f004:**
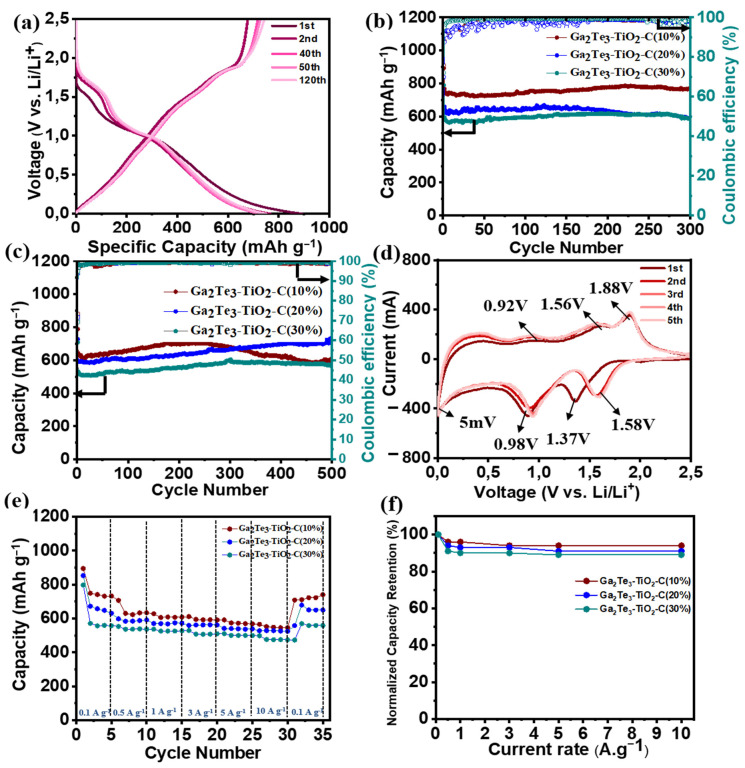
Electrochemical performance of Ga_2_Te_3_-TiO_2_-C composites for LIBs: (**a**) GCD curves of Ga_2_Te_3_-TiO_2_-C (10%) at 100 mA g^−1^; (**b**) cycling performance of Ga_2_Te_3_-TiO_2_-C composites at 100 mA g^−1^ and (**c**) 500 mA g^−1^; (**d**) CV curves of Ga_2_Te_3_-TiO_2_-C (10%); (**e**) rate capabilities of Ga_2_Te_3_-TiO_2_-C composites; and (**f**) capacity retention of Ga_2_Te_3_-TiO_2_-C composites from 0.1 to 10 A g^−1^.

**Figure 5 nanomaterials-12-03362-f005:**
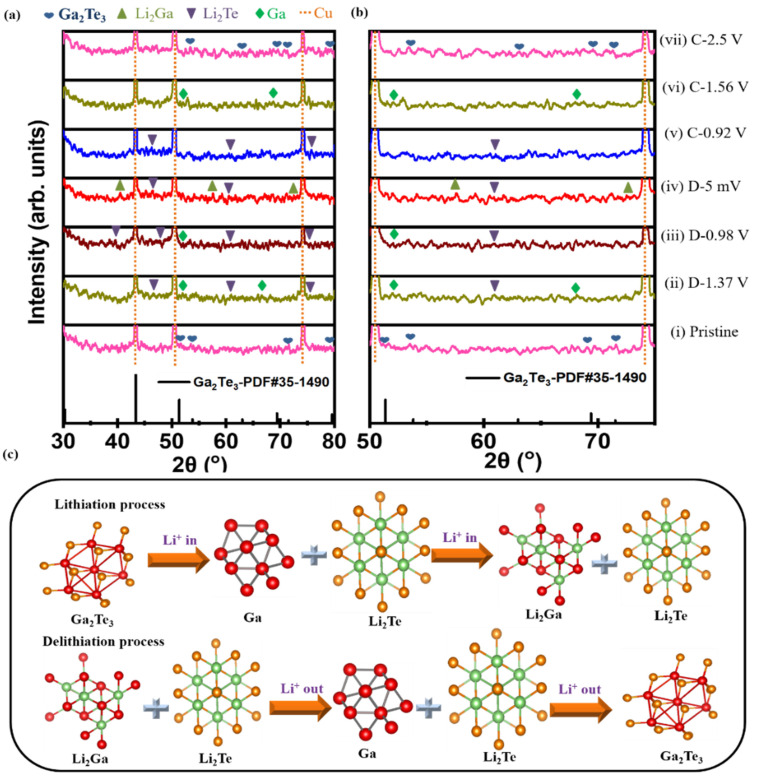
(**a**,**b**) Ex situ XRD patterns obtained at selected cutoff potentials in the initial discharge/charge process, and (**c**) schematics of phase change of Ga_2_Te_3_-TiO_2_-C (10%) electrode during cycling.

**Figure 6 nanomaterials-12-03362-f006:**
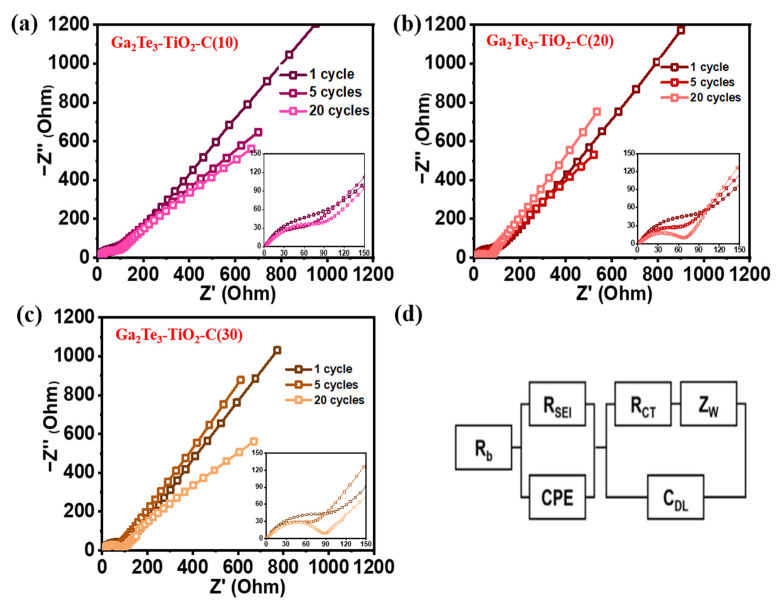
EIS-based Nyquist plots for (**a**) Ga_2_Te_3_-TiO_2_-C (10%), (**b**) Ga_2_Te_3_-TiO_2_-C (20%), (**c**) Ga_2_Te_3_-TiO_2_-C (30%) after 1, 5, and 20 cycles; and (**d**) equivalent circuit.

**Figure 7 nanomaterials-12-03362-f007:**
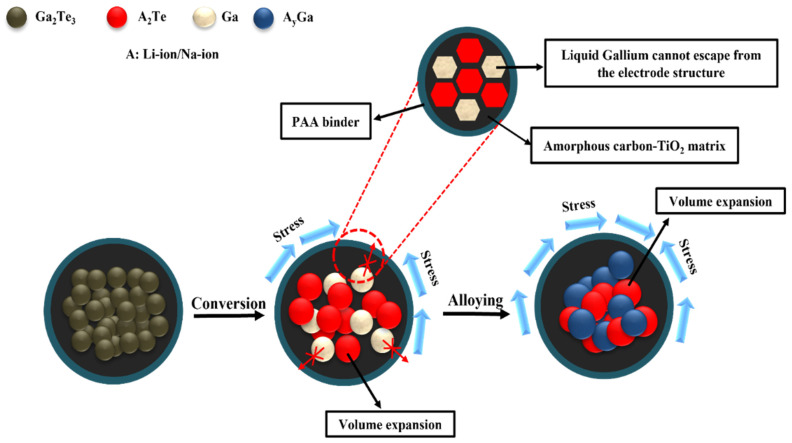
Schematic of reaction mechanism of Ga_2_Te_3_-TiO_2_-C (10%).

**Table 1 nanomaterials-12-03362-t001:** Performances of Ga-based intermetallic electrode for LIBs.

Anode	Cycling Performance	Rate Capability	Synthesis Method	Ref.
GaN-CNFs	405 mAh g^−1^ after 1200 cycles at 3 A g^−1^	310 mAh g^−1^ at 5 A g^−1^	Electrospinning	[95]
α-Ga_2_O_3_@G	350 mAh g^−1^ after 50 cycles at 0.15 A g^−1^	344 mAh g^−1^ at 0.5 A g^−1^	Hydrothermal and sintering process	[96]
Ga_2_O_3_/rGO	411 mAh g^−1^ after 600 cycles at 1 A g^−1^	222 mAh g^−1^ at 2 A g^−1^	Sol–gel method	[97]
Ga_2_O_3_/C	542 mAh g^−1^ after 200 cycles at 1 A g^−1^	192 mAh g^−1^ at 5 A g^−1^	One-step hydrogen reduction	[98]
Ga-Ni	420 mAh g^−1^ after 500 cycles at 3 C	410 mAh g^−1^ at 5C	Heating process	[99]
CuGa_2_	510 mAh g^−1^ after 65 cycles at 2 A g^−1^	440 mAh g^−1^ at 4 A g^−1^	Painting liquid Ga onto Cu foil	[100]
GaN/G	600 mAh g^−1^ after 1000 cycles at 1 A g^−1^	200 mAh g^−1^ at 10 A g^−1^	Wet chemical method	[101]
Ga_2_O_3_ NPs	721 mAh g^−1^ after 200 cycles at 0.5 A g^−1^	280 mAh g^−1^ at 2 A g^−1^	Hydrothermal carbonization method	[102]
Ga_2_S_3_	400 mAh g^−1^ after 20 cycles at 0.1 A g^−1^	-	Commercial material	[103,104]
SWCNT-GaS_x_	590 mAh g^−1^ after 100 cycles at 0.6 A g^−1^	-	Atomic layer deposition	[105]
GaSe	760 mAh g^−1^ after 50 cycles at 0.1 A g^−1^	450 mAh g^−1^ at 5 A g^−1^	Chemical reduction method	[106]
Ball-milled Ga_2_Te_3_/C	590 mAh g^−1^ after 500 cycles at 0.1 A g^−1^	495 mAh g^−1^ at 1C	Ball milling	[107]
Ga_2_Te_3_-TiO_2_-C	769 mAh g^−1^ after 300 cycles at 0.1 A g^−1^	600 mAh g^−1^ at 10 Ag^−1^	Ball milling	This work

## Data Availability

The data is available on reasonable request from the corresponding author.

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
