# Peer review of "Gallium-Telluride-Based Composite as Promising Lithium Storage Material"

_nanomaterials, 2022, doi:10.3390/nano12193362_

Round 1

Reviewer 1 Report

Ga2Te3-TiO2-C was successfully prepared via HEBM and investigated as a propitious anode material for LIBs in the text. The research offers some ideas to develop novel electrode material for lithium-ion batteries with high energy density. The text can be accepted by the journal of Nanomaterials after major revision.

1.The discharge capacity of the Ga2Te3-TiO2-C (10%, 20 and 30%) anodes for LIBs in rate test and long-cycle test is inconsistent at same current density. The authors should re-detect the two performances (rate and long cycle).

2. Please give the capacity contribution of C and TiO2 in the composites of Ga2Te3-TiO2-C (10%, 20 and 30%). And then give the real capacity of the Ga2Te3.

Reviewer 2 Report

The manuscript demonstrates a Ga2Te3-TiO2-C nanocomposite electrode to avoid liquid agglomeration of Ga, facilitate Li-ion diffusion kinetics, and accommodate volume change of the electrode by the TiO2-C matrix. Good rate capability and stable cycling performance are demonstrated. This will be of interest to the battery community. However, revisions are recommended before publication:

·         Please correct the mistakes in the statement “Among the chalcogenide elements, S and Se have been widely selected as anode materials in rechargeable LIB systems (e.g., Li-Se, Li-S)”. For Li-S and Li-Se batteries, S and Se are the cathode active materials rather than anode materials. Additionally, LIB usually refer to the Li-ion battery containing intercalation-type electrodes; however, Li-S and Li-Se batteries have conversion-type S and Se cathodes and require Li metal anodes or other Li-containing anodes, they are not usually considered as conventional LIBs.

·         While the introduction section introduces the background of alloy anodes, the conceptual inspiration of the Ga2Te3-TiO2-C composite electrodes is not provided. Please also include introduction to TiO2 and C, which are also very popular anode materials. The types of C should be introduced because different types of carbon (e.g., graphite, hard carbon, graphene, carbon nanotubes, carbon nanofiber, ect.) tend to have distinguished performance and functions.

·         What is the solvent for the electrode casting slurry? What is the solid-to-liquid ratio?

·         Are the current densities calculated based on per gram Ga2Te3-TiO2-C composite or per gram Ga2Te3? Please specify in the experimental section.

·         For Figure 2a, why are the TiO2 related XRD peaks below 20o missing from the XRD pattern of  Ga2Te3-TiO2?

·         Typo: In the description “the existence of Ti-O bonding with signals at 576.0 and 586.4 eV (Fig. 2d) on the Ga2Te3-TiO2-C(10%) surface implied that partial surface oxidation of active Ga2Te3”, Ti-O should be Te-O. In addition, since obvious oxidation is observed for Ga2Te3, is the composite anode air sensitive? Why is oxidation observed for Te but not for Ga? How would the oxidation of Ga2Te3 affect the electrode performance for practical use?

·         For Figure 2g, there will be adventitious carbon signal for XPS even if your sample does not contain C. How do you distinguish between the C 1s signal from you sample and the adventitious carbon?

·         Careful verification and revision of Figure 5a and 5b are required: the reference pattern should be labelled (seems to be Ga2Te3); the marked orange lines should be labelled, and dashed lines would be better than solid lines to avoid sight blocking of the XRD peaks; the labels for Ga2Te3, Li2Ga, Li2Te, and Ga seem to be completely off-position, as the labeled positions look like noise rather than peaks.

Round 2

Reviewer 1 Report

The revised text can be accepted by the journal of Nanomaterials.